# Effects of *Abelmoschus manihot* Flower Extract on Enhancing Sexual Arousal and Reproductive Performance in Zebrafish

**DOI:** 10.3390/molecules27072218

**Published:** 2022-03-29

**Authors:** Chi-Chang Chang, Jer-Yiing Houng, Wei-Hao Peng, Tien-Wei Yeh, Yun-Ya Wang, Ya-Ling Chen, Tzu-Hsien Chang, Wei-Chin Hung, Teng-Hung Yu

**Affiliations:** 1School of Medicine for International Students, College of Medicine, I-Shou University, Kaohsiung 82445, Taiwan; pengweihao@isu.edu.tw; 2Department of Obstetrics & Gynecology, E-Da Hospital/E-Da Dachang Hospital, Kaohsiung 82445, Taiwan; igiolal2011@gmail.com (Y.-L.C.); andy3560133@gmail.com (T.-H.C.); 3Department of Nutrition, I-Shou University, Kaohsiung 82445, Taiwan; jyhoung@isu.edu.tw; 4Department of Chemical Engineering, I-Shou University, Kaohsiung 82445, Taiwan; 5School of Chinese Medicine for Post-Baccalaureate, College of Medicine, I-Shou University, Kaohsiung 82445, Taiwan; isu10956013a@isu.edu.tw (T.-W.Y.); isu10956016a@isu.edu.tw (Y.-Y.W.); 6School of Medicine, College of Medicine, I-Shou University, Kaohsiung 82445, Taiwan; ed102600@edah.org.tw (W.-C.H.); ed102599@edah.org.tw (T.-H.Y.); 7Division of Cardiology, Department of Internal Medicine, E-Da Hospital, Kaohsiung 82445, Taiwan

**Keywords:** *Abelmoschus manihot*, sexual arousal, reproductive performance, gene expression, zebrafish

## Abstract

The flower of *Abelmoschus manihot* L. is mainly used for the treatment of chronic kidney diseases, and has been reported to have bioactivities such as antioxidant, anti-inflammatory, antiviral, and antidepressant activities. This study used wild-type adult zebrafish as an animal model to elucidate the potential bioactivity of *A. manihot* flower ethanol extract (AME) in enhancing their sexual and reproductive functions. Zebrafish were fed AME twice a day at doses of 0.2%, 1%, and 10% for 28 days, and were then given the normal feed for an additional 14 days. The hormone 17-β estradiol was used as the positive control. Sexual behavioral parameters such as the number of times males chased female fish, the production of fertilized eggs, and the hatching rate of the fertilized eggs were recorded at days 0.33, 7, 14, 21, 28, and 42. The expression levels of sex-related genes—including *lhcgr*, *ar*, *cyp19a1a*, and *cyp19a1b*—were also examined. The results showed that the chasing number, fertilized egg production, and hatching rate were all increased with the increase in the AME treatment dose and treatment time. After feeding with 1% and 10% AME for 28 days, the chasing number in the treated group as compared to the control group increased by 1.52 times and 1.64 times, respectively; the yield of fertilized eggs increased by 1.59 times and 2.31 times, respectively; and the hatching rate increased by 1.26 times and 1.69 times, respectively. All three parameters exhibited strong linear correlations with one another (*p* < 0.001). The expression of all four genes was also upregulated with increasing AME dose and treatment duration. When feeding with 0.2%, 1%, and 10% AME for 28 days, the four sex-related genes were upregulated at ranges of 1.79–2.08-fold, 2.74–3.73-fold, and 3.30–4.66-fold, respectively. Furthermore, the effect of AME was persistent, as the promotion effect continued after the treatment was stopped for at least two weeks. The present findings suggest that AME can enhance the endocrine system and may improve libido and reproductive performance in zebrafish.

## 1. Introduction

Sexual health is of great significance to personal health and quality of life for both men and women. It is important for people to have a satisfying sex life for their work performance and wellbeing. In addition to improving the quality of life, passionate sex can also prevent the degeneration of sexual organs and gonads, thereby maintaining endocrine balance.

Low sexual desire affects more than 20% of women, and increases with age [1,2]. The most common sexual issues in women are lack of sexual interest, dyspareunia, inability to achieve orgasm, unsatisfactory sexual relationships, and women who are postmenopausal [3,4]. In addition, hyposexuality disorder is also often associated with depression, anxiety, and negative emotional and psychosocial factors, such as marital conflict, partner sexual dysfunction, life stress, and religious and social group customs [5,6]. Such dysfunction is usually related to decreases in sex hormone (estrogens and androgens) concentrations. Some studies have reported that decreased sexual activity in postmenopausal women is associated with decreased testosterone levels [7,8]. Treatment with estradiol can increase the libido of postmenopausal women by acting on the central nervous system [9]. Estradiol also acts indirectly on the vaginal wall to increase lubrication and reduce sexual disturbances [10].

For men, loss of libido and lack of sexual satisfaction are considered the most common types of sexual dysfunction [4,11]. About 2.3–7.7% of men (dependent on age) in Denmark [12] and 15% of men in Canada [1] have low-sexual-desire problems. This could be caused by low testosterone, which is the main hormone that regulates men’s sexual desire [13]. Plasma testosterone concentrations can be affected by a number of comorbidities, including erectile dysfunction, premature ejaculation, diabetes, hypertension, depression, and lower urinary tract symptoms [14,15], as well as lifestyle factors such as obesity, life stress, and insufficient physical activity [16,17].

Nutritional interventions, such as dietary supplements, have received increasing attention in improving sexual and reproductive function in males and females. Several medicinal plants—such as *Avena sativa*, *Eurycoma longifolia*, *Ginkgo biloba*, *Psoralea corylifolia*, *Tribulus terrestris*, and *Withania somnifera*—have been used to treat sexual dysfunction or to promote sexual desire and reproductive behavior [18,19,20].

*Abelmoschus manihot* (L.) Medicus (Malvaceae) is a perennial shrub widely distributed in China, India, and Taiwan. The flower of this plant, rich in flavonoids, has been used in China mainly to treat chronic kidney diseases [21,22,23] and diabetic nephropathy [24,25,26]. In addition, the extract of the *A.*
*manihot* flower has been demonstrated to possess antiviral [27], antidepressant [28,29], anti-adipogenic [30], antioxidant [30,31], hepatoprotective [32], pro-angiogenic [33,34], anti-triglyceride accumulation [35], and anti-inflammatory activities against inflammatory gut diseases [36], along with cardioprotective action against myocardial ischemia [37,38,39], and the ability to improve bone care [40,41]. It is an old saying from folklore that drinking *A.*
*manihot* flower tea can enhance human sexual desire and performance. However, until today, no related scientific evidence has been reported to support this claim.

Zebrafish (*Danio rerio*) have a high degree of homology with the human genome and well-characterized gene functions, increasing the credibility and applicability of our research findings [42]. Recently, zebrafish have become the second most widely used vertebrate model in human biomedical and functional genomics research [43,44]. In addition to their pharmacological and physiological similarities with mammals, zebrafish have several additional advantages, including small embryo and larval size, optical transparency, rapid development, short reproductive cycle, and high fecundity. The use of zebrafish to replace mice or rats is also in line with the trend of the times—that is, using smaller animals to replace larger animals as a research model [19,20,45,46]. To date, zebrafish have also become a reliable model organism for reproductive and infertility assessment, as well as for studies of pair-mating behavior [47,48,49,50,51]. Accordingly, we used zebrafish as a model to evaluate whether the *A.*
*manihot* flower indeed has promotive effects on sexual performance. In this study, the chasing number, fertility, hatching rate, and the expression of the related genes in the brain and gonads in zebrafish treated with *A.*
*manihot* flower ethanol extract (AME) were examined.

## 2. Results

### 2.1. Effects on Sexual Performance

#### 2.1.1. Chasing Behavior

Separately housed male and female zebrafish were given different extract-added diets twice daily for a period of 28 days. The hormone 17-β estradiol (1.8 nM) was used as the positive control. All fish were then switched to a normal fish diet to examine whether the enhancement of sexual performance could be maintained after feeding with the extract was discontinued. At the beginning of the dark cycle on the 8th hour (first day) and the 7th, 14th, 21st, 28th, and 42nd days after feeding on the normal diet, one female and one male fish were acclimated in different chambers of the mating tank, separated with a transparent plastic divider and permitted water flowing across the divider. The barrier was removed at the beginning of the daylight period on the next morning. This allowed male and female zebrafish to approach one another. We used a camera to record all interactions for the next 30 min, and counted the chasing number between the two fish.

As shown in Figure 1, AME promoted the chasing response between male and female zebrafish in a dose-dependent manner (correlation coefficient R = 0.67, *p* < 0.05 at day 28). Those supplemented with low-dose AME (0.2%) did not show a significant promotive effect, even up to 28 days. Those fed with 1% AME showed a significant effect (*p* < 0.05) starting at 8 h. The effect on those fed with 10% AME was close to that of the positive control group (estradiol). Furthermore, the longer the AME was administered, the more mating occurred (R = 0.96–0.99, *p* < 0.001 for 1% and 10% AME doses).

#### 2.1.2. Spawning Outcome

After 30 min of chasing, the fish in the mating tank were moved out, and the fertilized eggs that fell on the bottom of the incubator were collected and counted. The numbers of embryos observed at various times after feeding with AME are shown in Figure 2. The results show that AME supplementation increased the number of fertilized eggs, in direct association with the AME treatment dose (R = 0.92, *p* < 0.001 at day 28) and feeding time (R = 0.97, *p* < 0.001 for 1% and 10% AME doses).

The effect of feeding with AME on the hatching rate was determined from the percentage of hatched embryos among the total viable embryos. The number of embryos that hatched normally was recorded after placing the fertilized eggs in the incubator for 60 h post-fertilization (hpf). The results in Figure 3 illustrate that feeding with more than 1% AME had a significant promoting effect on the hatching rate. The increase was related to the increase in the treatment dose (R = 0.89, *p* < 0.01 at day 28) and the extension of the feeding time (R = 0.68–0.77, *p* < 0.01 for 1% and 10% AME doses).

#### 2.1.3. Correlations between Mating and Spawning Outcomes

The correlations between the parameters of fertilized egg production and chasing number, hatching rate and chasing number, and hatching rate and fertilized egg production were R = 0.98 (*p* < 0.001), R = 0.97 (*p* < 0.001), and R = 0.98 (*p* < 0.001), respectively. These results demonstrate that AME administration stimulated the mating between male and female zebrafish, and subsequently increased the fertilized egg production and hatching rate.

#### 2.1.4. Body Weight

Figure 4 shows the effects of AME supplementation for 28 days on zebrafish body weight. Supplementation with AME had no significant effect on body weight in male or female fish.

#### 2.1.5. Lasting Effect of AME

In order to investigate the sustainability of the effect of AME on zebrafish’s sexual performance, the above treatment experiments were conducted for 28 days, twice a day, and switched to the normal fish diet until the 42nd day. The results show that the chasing behavior between male and female fish (Figure 1), along with the egg production (Figure 2), remained significantly different from those of the vehicle group on day 42. In terms of the hatching rate of the fertilized eggs, all AME groups and estradiol group showed a slight downward trend (Figure 3). Nonetheless, the 10% AME group and the estradiol group remained significantly different from the vehicle group on day 42 (*p* < 0.01).

### 2.2. Effects on the Expression of Related Genes

The effects of feeding with AME on the expression of several genes related to sexual development, sexual behavior, and reproduction—such as *lhcgr*, *ar*, *cyp19a1a*, and *cyp19a1b*—were analyzed using the RT-PCR method. Zebrafish are a gonochoristic species with two sexes. During the sexual differentiation of juvenile fish, their reproductive organs can undergo sexual reversal due to changes in their environment, such as the presence of sex steroids. However, their sex does not change after maturing into an adult fish. Therefore, both male and female mature fish will have these sex-related genes. These genes are present in many different organs or tissues in the fish. In order to examine the effects of *A. manihot* extracts on the expression of these sex-related genes in male and female fish, we examined the genes based on their predominant functions in male or female fish and their main organ tissues. In this study, *lhcgr* in the ovaries and *cyp19a1a* in the brains of female fish, along with *ar* in the testis tissue and *cyp19a1b* in the brains of male fish, were investigated.

Figure 5 shows the effects of feeding with AME on the gene expression in female and male fish. Even when feeding with AME at concentrations as low as 0.2%, the expression of these sex-related genes was enhanced. The levels of expression of all four of these genes increased with the increase in the dosage and the feeding time (until day 28). After 28 days of feeding with AME, all fish were changed to a normal diet. The effect of AME on the expression of these genes could still be found at day 42, indicating that AME has a persistent effect on fish.

### 2.3. Chemical Composition of AME

Flavonoids are known to be some of the main chemical constituents of *A. manihot* flower extract [52,53]. In this study, the flavonoid compounds of AME were analyzed via high-performance liquid chromatography (HPLC), and the chromatograph is shown in Figure 6. Five components were identified: rutin, hyperoside, isoquercitrin, myricetin, and quercetin. Table 1 demonstrates that the contents of hyperoside and myricetin were high in AME, followed by isoquercitrin, rutin, and quercetin. The total polyphenol content (TPC) and total flavonoid content (TFC) in AME were also detected, and are listed in Table 1.

## 3. Discussion

Sexual health is important for overall health, quality of life, and the continuation of future generations. Sexual activities can also activate the gonads and sexual organs so as to maintain the homeostasis of human endocrines. Low sexual desire and arousal dysfunction are considered the most common problems of sexual dysfunction in men and women [3,4,11]. Sexual and reproductive health is highly correlated with daily diet, so focusing on diet and consuming specific dietary supplements—such as herbal extracts and natural products—is an effective way to improve sexual health [11,54].

*A. manihot* flower, in traditional medicine, has the effect of promoting sexual desire. This study conducted the preliminary verification of this potential effect using a wild zebrafish model. To our knowledge, this is the first report describing the effect of *A. manihot* flower extract on enhancing sexual performance.

Zebrafish are often used as an animal model to investigate the effects of specific substances—including hormones, toxic substances, and sex-related genes—on sexual behavior and reproductive function [42,48,55,56,57,58]. Zebrafish have also become a promising model for assessing sexual and reproductive performance, due to advances in the integrity of courtship behavior, developmental and physiological studies, and their short reproductive cycle. Furthermore, the high similarity of reproductive regulatory systems between humans and zebrafish allows researchers to use zebrafish as a model to study reproduction-related topics in a more comprehensive manner [42,59,60].

In this study, the mating behavior of male and female zebrafish was used to observe the arousal of sexual desire. Mating behavior in zebrafish includes male chasing, lateral contact preceding egg-laying, and the release and fertilization of eggs. The hatching rate value is used to represent the health of reproductive capacity [61]. The results of this study show that AME significantly increased the chasing number between male and female fish, the number of fertilized eggs, and the hatching rate, in a dose- and treatment-time-dependent manner (Figure 1, Figure 2 and Figure 3). The significant increase in chasing number indicates that AME has an arousing effect on libido. The increase in the number of embryos indicates that AME is effective in enhancing the mating between male and female fish. Furthermore, the increase in the hatching rate indicates that AME promotes healthy reproduction.

The effect of AME on sexual behavior appears to last for a certain length of time, as high sexual performance can still be observed—but is decreased somewhat—in fish after stopping feeding with the AME for 14 days. The treatment with AME did not significantly change the body weight of the fish (Figure 4). Since no zebrafish died throughout the whole experiment, this proved that consuming AME is safe.

The mechanism underlining how AME affects sexual behavior in zebrafish may exert its effect on sex hormone activity. In males, androgens are the male sex hormones essential for the development and maintenance of the reproductive system. The androgen receptor (AR) regulates the expression of androgen-related genes, which are crucial in the male reproductive system—especially spermatogenesis. Since AR acts as a major regulator of androgen-related signaling pathways, dysfunction of androgen/AR signaling interferes with the reproductive system’s development, including the male reproductive tract, gonadal development, and sexual behavior [62,63].

In females, the luteinizing hormone/choriogonadotropin receptor (LHCGR)—a transmembrane receptor found mainly in the ovaries—is expressed in several cells, such as luteal cells, thecal cells, stromal cells, and granulosa cells. It is involved in processes such as follicle maturation, ovulation, and luteal function. The receptor interacts with luteinizing hormone (LH) and chorionic gonadotropin, and is required for hormonal function during reproductive process. Signals from LHCGR are required for steroidogenesis and gametogenesis in female fish. Thus, dysfunction of the LHCGR in females can lead to infertility [64,65].

It is known that the production of sex hormones is regulated by the expression of their related genes. For example, the final step in estrogen production is catalyzed by cyp19 aromatase in mammals, which converts C19 androgens (such as testosterone or androstenedione) into C18 estrogens (such as 17β-estradiol or estrone). The genome of most teleost fish—including zebrafish—contains two cyp19 aromatase genes with different structures: the *cyp19a1a* gene encoding aromatase A, and the *cyp19a1b* gene encoding aromatase B. Adult fish have high aromatase activities in the brain [66,67]. Thus, gene expression of cyp19 aromatase affects androgen conversion to estrogen, maturation of gonads, and normal maintenance of sexual behavior [68,69].

The results in this study show that AME administration could upregulate the *lhcgr* and *cyp19a1a* genes in female fish, as well as the *ar* and *cyp19a1b* genes in male fish—all in a dose- and time-dependent manner (Figure 5). This implies that the AME can enhance sexual behavior by promoting the expression of these genes in the brain and the gonads, and can also elevate the endocrine system. Therefore, AME exerts its positive effects on sexual arousal and reproductive performance through regulating the expression of sex-related genes.

The chemical composition of *A. manihot* flower extract mainly contains flavonoids, amino acids, nucleosides, organic acids, steroids, polysaccharides, and volatile compounds. Many studies have reported that the major pharmacologically active components in *A. manihot* flower are flavonoids [22,52,53]. Luan et al. [52] described 49 flavonoids isolated and identified from the flowers of *A. manihot*. Among them, hyperoside, myricetin, hibifolin, isoquercetin, quercetin-3-O-robinobioside, rutin, and quercetin are considered to be the main bioactive components [70,71]. In this study, we identified five flavonoids in AME via HPLC, as shown in Figure 6 and Table 1. The contents of these flavonoids in AME were similar to the data reported in the literature.

Accumulated studies have demonstrated that flavonoids and phenolic compounds exhibit a wide range of biological activities, such as antioxidant, anti-inflammatory, anticancer, anti-obesity, antidiabetic, and neuroprotective effects [72,73,74]. It has been shown that some polyphenols and flavonoids can cross the blood–brain barrier and exhibit neuropharmacological effects, including inhibition of catechol-O-methyltransferase (COMT) and monoamine oxidase (MAO) activities, leading to an increase in dopamine concentration in the medial preoptic area (MPOA) of the brain, thus resulting in arousal of sexual instincts, desire, and motivation [19,75,76,77,78,79].

It is also known that excess reactive oxygen species (ROS) can generate oxidative stress, causing damage to sperm proteins, lipids, and DNA, leading to sperm dysfunction—including sperm motility issues—DNA alterations, and reduced membrane integrity [80]. Flavonoids and phenolic compounds tend to have excellent antioxidant activities that can effectively scavenge free radicals and reduce oxidative stress, so they may have the potential to protect sperm and improve pregnancy rates. In this study, all of the beneficial effects were observed in fish fed the total extract of the flower.

Since the total extract contains many other compounds in addition to flavonoids and polyphenols, whether or not these compounds exert any sexual-function-promoting effect merits further investigations.

## 4. Materials and Methods

### 4.1. Preparation of AME

The *A. manihot* (L.) Medicus plant materials were purchased from the Kangmei Traditional Chinese Medicine Store (Bozhou City, Anhui, China). Its nucleotide sequence had been determined (Appendix A) and submitted to GenBank (http://www.ncbi.nlm.nih.gov/nucleotide (accessed on 3 February 2021)). The closest match resulting from the BLASTN search was Sequence ID: KY218782.1 (23 to 722 bp), with 99.43% identity. For the preparation of *A. manihot* flower ethanol extract (AME), 2.6 kg of dried flowers was first crushed and pulverized. The powder was then extracted with 16 L of 95% ethanol overnight. The extracted solution was filtered and collected. The remaining residue was extracted twice with 16 L of 95% ethanol. The extracted solution was combined and evaporated to remove the solvent using a vacuum evaporator (Panchum Scientific Co., Kaohsiung, Taiwan). The extract residue was further dried by lyophilization in a freeze-dryer (Panchum Scientific Co.). The total dry mass of AME obtained was 654 g, so the yield was 25.2%. The dried residue was stored at −20 °C before use.

### 4.2. Fish Husbandry

The protocol of experimental procedures was approved by the Institutional Animal Care and Use Committee of I-Shou University (AUP-107-43-01), in accordance with the regulations of the local and central government.

The wild-type AB strain adult zebrafish were purchased from a commercial breeder (Kaohsiung, Taiwan). Prior to the experiment, all fish were acclimated to the laboratory environment and operational conditions for at least 2 weeks. Fish were reared at 28 °C under a light/dark photoperiod of 14/10 h. Water quality was maintained by circulation using a Fluval U2 aquarium internal filtration system (Rolf C. Hagen Inc., Quebec, QC, Canada). One-third of the water was changed daily. Fish were fed twice, with a fixed amount of 0.025 g/fish/meal per day.

Approximately three-month-old adult fish were randomly allotted to different glass aquariums (50 × 30 × 30 cm^3^). The number of fish used in each group of experiments was as stated in each experimental method. For each study, five replicates of independent experiments were performed.

### 4.3. Sexual Behavior Evaluation

The AME was blended with a commercial fish feed (Talkong Inc., Tainan, Taiwan) at 3 different ratios: 0.2% extract diet (1:499), 1% extract diet (1:99), and 10% extract diet (1:9). All diets were air-dried, placed in airtight containers, and stored in a –20 °C refrigerator until use.

Adult male and female fish were assigned separately into different glass aquariums (50 × 30 × 30 cm^3^), with a maximum density of 30 fish per tank, and were fed with 0, 0.2, 1, and 10% extract diets, twice per day, for 28 days. The 17-β estradiol of 1.8 nM was used as the positive control group. After that, fish were fed with the normal diet for 14 more days until the end of feeding (the 42nd day). During the feeding period, one-third of the water was replaced with fresh water daily. At 8 h on the first day, and the 7th, 14th, 21st, 28th, and 42nd days after feeding, a female fish and a male fish were picked up separately and acclimated in a chamber of the mating tank with a transparent plastic divider at beginning of the dark cycle. The next morning, at the start of the light cycle, the barriers were removed, allowing the male and female zebrafish to approach one another. All interactions over the next 30 min were recorded on a video camera (SJ8 PRO model, SJCAM, Shenzhen, Guangdong, China). Two behaviors—chasing and spawning—were assessed. The number of chasing events was estimated as the number of times per minute that the male directly and aggressively swam towards the female, causing it to increase speed.

When the chasing behavior ceased and the female zebrafish spawned, the fish were removed and the eggs that fell to the bottom of the tank were collected. The fertilized eggs were screened and their number was counted, and then they were transferred to sterile glass Petri dishes containing egg water. The Petri dishes were placed in a 28 °C incubator (Firstek Scientific Co., New Taipei City, Taiwan), and the number of embryos hatched at 60 h post-fertilization (hpf) was recorded. The hatching rate represents the percentage of hatched embryos among the total living embryos. The same experiment was repeated five times at each sampling time point for each dietary treatment.

The fish that were taken out at each sampling time point were not returned to the original incubator after the above experiments. In other words, these fish were not reused for other experiments.

### 4.4. Gene Expression Analysis with Quantitative RT-PCR

As described by Pradhan and Olsson [57], the brain and gonad tissue (ovary or testis) samples of female and male zebrafish were taken and kept at −20 °C until use.

RNA isolation was performed using the Qiagen RNeasy Kit (Qiagen AG, Venlo, the Netherlands), according to the manufacturer’s protocol. The mRNA was reverse transcribed to cDNA using the Magic RT cDNA synthesis kit (Bio-Genesis, Taipei, Taiwan). The obtained cDNA was then amplified using the IQ2 SYBR Green Fast qPCR System Master Mix LOW ROX kit (Bio-Genesis) with the QuantStudio™ 3 qPCR System (Thermo Fisher Scientific, Waltham, MA, USA). The qPCR reactions began with a holding stage for 2 min at 50 °C, followed by another holding stage of 10 min at 95 °C. A total of 40 cycles were run (first 15 s each at 95 °C, followed by 1 min at 60 °C). The specific primers used for qRT-PCR analysis are listed in Table 2. Amplification of β-actin mRNA was used as an internal control. Each experiment was performed five times, and each sample was detected in triplicate. Following the PCR procedure, qRT-PCR data were analyzed with IQ5 Optical System Software (Bio-Rad, Hercules, CA, USA), the baselines of which were automatically set by the software.

### 4.5. Analysis of Chemical Compositions by HPLC

The AME was dissolved in methanol and then filtered through a 0.22 μm membrane filter. The contents of flavonoids in AME were analyzed with an HPLC system (LC20-AT model, Shimadzu, Kyoto, Japan) equipped with a C18 column (250 mm × 4.6 mm, 5 μm; Supelco, Bellefonte, PA, USA). The mobile-phase solution consisted of solvent A (acetonitrile) and solvent B (0.1% acetic acid in water) for gradient adjustment. The program for gradient elution consisted of 0–100 min (12–40% A) and 100–110 min (40–12% A). The flow rate was 1.0 mL/min, and the sample injection volume was 20 μL. Detection of eluates was performed at a wavelength of 320 nm. The compounds in the eluents were identified by comparing their retention times with reference compounds. The concentration of the eluates was quantified by interpolation from the standard curves constructed by linear regression using Microsoft Excel software (Microsoft Software Inc., Redmond, WA, USA). Flavonoid references such as rutin, hyperoside, isoquercitrin, myricetin, and quercetin were purchased from ChromaDex (Los Angeles, CA, USA). Naringin, obtained from Sigma-Aldrich (St. Louis, MO, USA), was used as the internal standard in this analysis.

The TPC and TFC of each extract were analyzed as described by Tsai et al. [81]. Briefly, for determination of TPC, the extract sample was mixed with Folin–Ciocalteu reagent (Sigma-Aldrich) and sodium carbonate solution. The mixture was allowed to stand at room temperature for 30 min, and its absorbance at 765 nm was detected with an ELISA reader (SPECTROstar Nano, BMG LABTECH, Offenburg, Germany). The content of TPC was calculated by interpolation from the calibration curve, which was established using gallic acid as the standard. The content of TPC in dry extract was expressed as gallic acid equivalents.

To determine the TFC, the extract sample was added to sodium carbonate solution and allowed to stand for 5 min. The aluminum chloride (AlCl_3_) was then poured into the solution and kept for another 6 min. After that, 0.5 mL of 1 M sodium hydroxide (NaOH) solution and 0.275 mL of deionized water were added. After mixing, the absorbance of the solution was detected at 510 nm. The concentration of TFC in the dry extract was calculated by interpolation from the calibration curve, using catechin as the standard, and expressed as catechin equivalents.

### 4.6. Statistical Analysis

All experiments were conducted five times. Data were expressed as the mean ± standard deviation (SD). Statistical differences between variants were analyzed by one-way ANOVA. The significance of experimental data and the correlation between different parameters were analyzed using SPSS 25.0 (SPSS Inc., Chicago, IL, USA). Levels of significance were defined as * *p* < 0.05, ** *p* < 0.01, and *** *p* < 0.001.

## 5. Conclusions

This study demonstrated that the administration of AME can enhance sexual arousal and reproductive performance in zebrafish without fatal cases and without affecting body weight. AME administration also upregulated the sex-related genes, such as *lhcgr*, *ar*, *cyp19a1a*, and *cyp19a1b*. Accordingly, this study has clearly shown that AME can enhance sexual function in zebrafish. However, since this study used a total extract that contained many other compounds, whether it is flavonoids, polyphenols, or other components that have this sexual-function-promoting effect, as well as is the nature of the mechanism underlying this effect, still need further investigations.

## Figures and Tables

**Figure 1 molecules-27-02218-f001:**
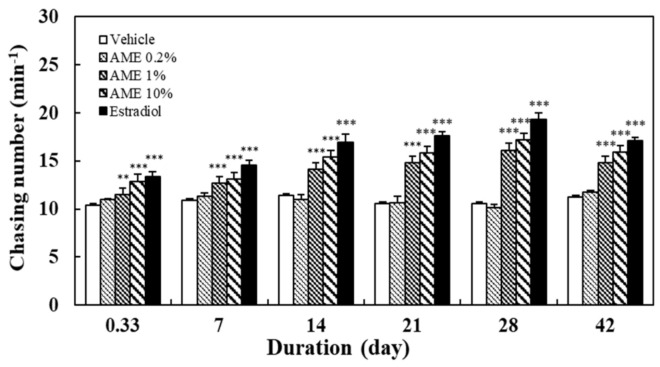
Effects of the supplementation with AME on the chasing number between the male and female zebrafish: Each dataset is represented by five independent repeated experiments. Statistical differences compared to the vehicle group were tested using one-way ANOVA. Levels of significance are expressed as ** *p* < 0.01 and *** *p* < 0.001.

**Figure 2 molecules-27-02218-f002:**
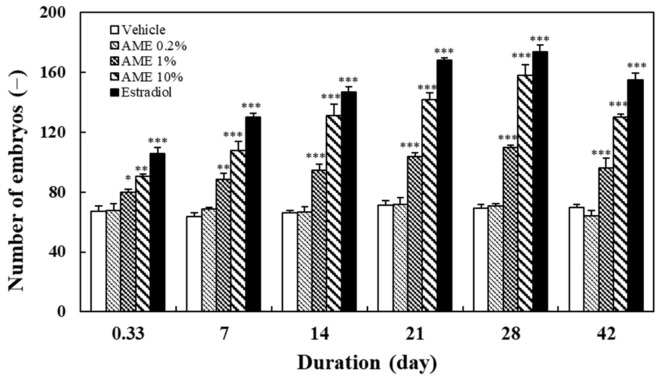
Effects of the supplementation of AME on the fertilized egg production. Each dataset is represented by five independent repeated experiments. Statistical differences compared to the vehicle group were tested using one-way ANOVA. Levels of significance are expressed as * *p* < 0.05, ** *p* < 0.01, and *** *p* < 0.001.

**Figure 3 molecules-27-02218-f003:**
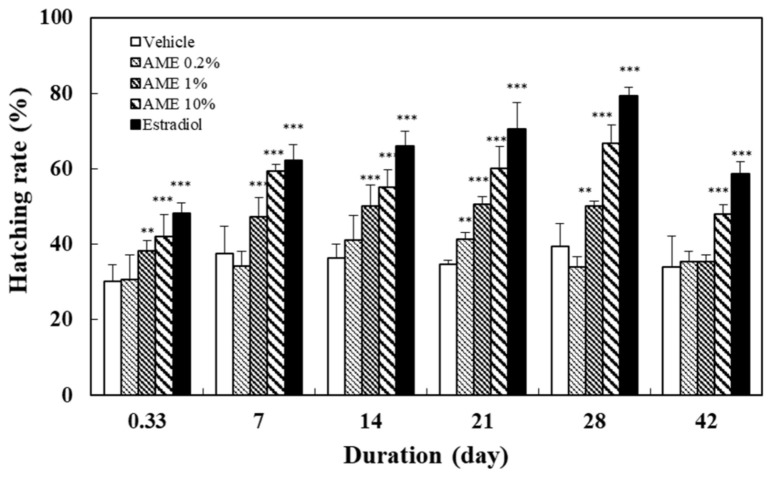
Effects of the supplementation with AME on the hatching rate of the fertilized eggs. Each dataset is represented by five independent repeated experiments. Statistical differences compared to the vehicle group were tested using one-way ANOVA. Levels of significance are expressed as ** *p* < 0.01 and *** *p* < 0.001.

**Figure 4 molecules-27-02218-f004:**
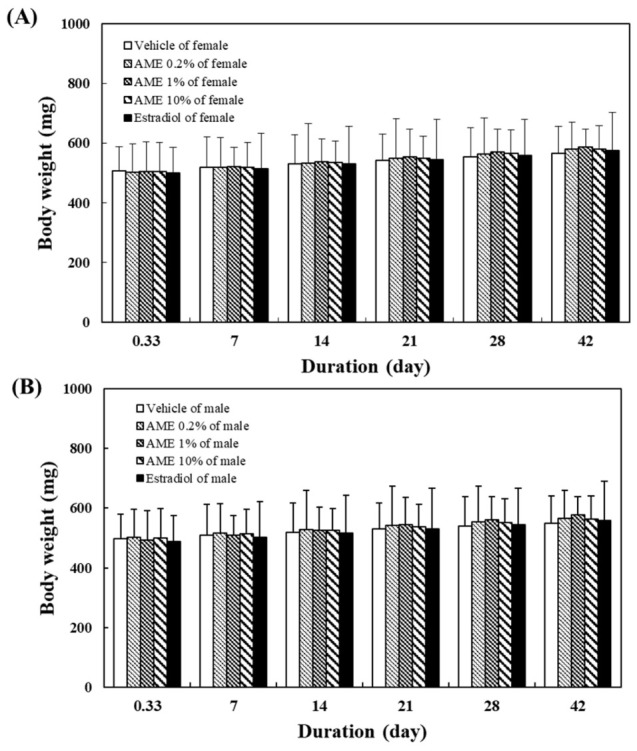
Effects of the supplementation with AME (**A**: Female, **B**: Male) on the body weight of zebrafish during 42-day cultivation. At each sampling time, five male or female adult fish were weighed.

**Figure 5 molecules-27-02218-f005:**
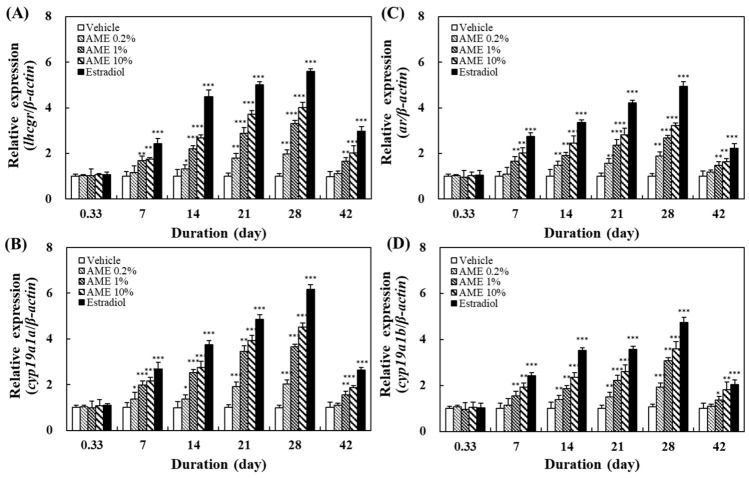
Effects of AME stimulation on the expression of sex-related genes in zebrafish: (**A**) female, *lhcgr* in ovaries; (**B**) female, *cyp19a1a* in the brain; (**C**) male, *ar* in the testes; (**D**) male, *cyp19a1b* in the brain. Five adult zebrafish were used for each experimental group at each sampling time. Statistical differences in gene expression between the AME-supplemented-diet group and the vehicle group were tested using one-way ANOVA, and levels of significance are denoted as * *p* < 0.05, ** *p* < 0.01, and *** *p* < 0.001.

**Figure 6 molecules-27-02218-f006:**
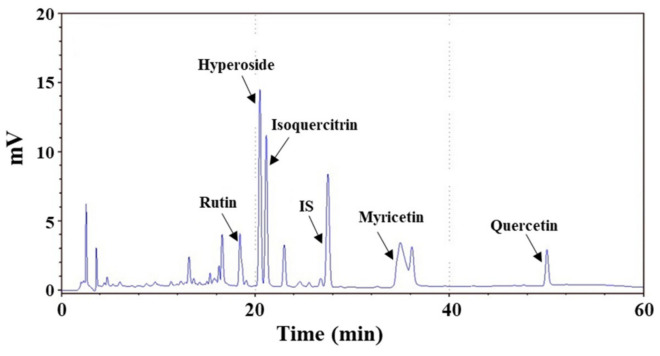
HPLC chromatograms of AME: Identified components—rutin, hyperoside, isoquercitrin, myricetin, and quercetin. The peak of the internal standard (naringin) is denoted as “IS”.

**Table 1 molecules-27-02218-t001:** Chemical compositions of AME.

Chemical Composition	Concentration (mg/g Extract)
Rutin	10.0 ± 0.4
Hyperoside	37.8 ± 2.1
Isoquercitrin	20.6 ± 1.7
Myricetin	36.9 ± 1.4
Quercetin	2.6 ± 0.1
Total polyphenols content	120.8 ± 0.8
Total flavonoids content	57.0 ± 1.0

**Table 2 molecules-27-02218-t002:** The primers to detect gene expression in RT-PCR analysis.

Primer	Sequence
β-Actin	5′-CACCATGAAGATCAAGATCA-3′
	5′-TTTATTCAAGATGGAGCCACCGATCC-3′
*lhcgr*	5′-GGCTGACCTGTCTGCAATCT-3′
	5′-GAAATAGGCGCCATGCACAG-3′
*ar*	5′-CGTAGGATGCACGTCTCCAG-3′
	5′-AGTCCATCAGTCGTGTCAGC-3′
*cyp19a1a*	5′-GGCACACGCAGAGAAACTTG-3′
	5′-GCTGGAAGAAACGACTCGGA-3′
*cyp19a1b*	5′-TTTCAGTACCTGGCAGACGG-3′
	5′-GTCAGCCGACTCTACGTCTC-3′

## Data Availability

The data presented in this study are available in this article.

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
