# Peer review of "Effects of Abelmoschus manihot Flower Extract on Enhancing Sexual Arousal and Reproductive Performance in Zebrafish"

_molecules, 2022, doi:10.3390/molecules27072218_

Round 1

Reviewer 1 Report

  1. The title is interesting and well written by the authors but still, lacks some issues like the abstract section is poorly written, it should contain a summary of the results and should exclude the introduction part from it. The abstract must provide enough quantitative information that will allow a reader to understand the key findings of the research.
  2. The introduction part of the manuscript is not written well which constitutes the lack of crux in it. The major part of the introduction consists of sexual health and is less focused on the plant and model used for this activity.
  3. The authors are suggested to submit an authentication letter of crude drug samples. As they purchased the flower of the plant from a store so it is hard to find the purity and originality.
  4. Please do explain why you have taken ethanol as a solvent for the extraction method?
  5. For the extraction the authors use ethanol that is proven mutagen and carcinogenic. So, they are advised to go for the residual solvent content of the extract.
  6. Please mention the manufacturer details of the standard diet and/ or its composition if available.
  7. How the authors can determine the dose for pharmacological activity? If it is already mentioned somewhere in the literature. Please cite in the text.
  8. Please mention the manufacturer details of the incubator and other instruments used throughout the manuscript.
  9. Authors are strictly advised to first complete the full form of words then later they can use abbreviations (AlCl3 at line number 397).
  10. The conclusion section is poorly written.
  11. The manuscript needs standard review.

Author Response

Dear Referee,

Attached please find our revised manuscript (molecules-1650301) entitled “Effects of Abelmoschus manihot flower extract on enhancing sexual arousal and reproductive performance in zebrafish” which we wish to publish in Molecules journal. We thank you for your thoughtful comments and constructive suggestions, which greatly improve the present manuscript. Following your comments and suggestions, we have carefully revised the manuscript, and listed our point-by-point responses as follows.

Q1.  The title is interesting and well written by the authors but still, lacks some issues like the abstract section is poorly written, it should contain a summary of the results and should exclude the introduction part from it. The abstract must provide enough quantitative information that will allow a reader to understand the key findings of the research.

Reply: Thanks for the comment. The abstract has been rewritten, parts of introductory remarks have been removed, and more quantitative data added.

Q2.  The introduction part of the manuscript is not written well which constitutes the lack of crux in it. The major part of the introduction consists of sexual health and is less focused on the plant and model used for this activity.

Reply:   Thanks for the comment. To the best of our knowledge, there is no previous study working on this kind of activity in this plant. Thanks for the reviewer's reminder: the statement on Line 97-99 was indeed not enough clear, so we have now made it clearer by moving it to Line 83-85. With respect to the less described animal models, we have added the commonly used rat and mouse models and their relevant literatures to Line 93-94 in the revised paper, Line 96-101 has also been revised as well.

Q3.    The authors are suggested to submit an authentication letter of crude drug samples. As they purchased the flower of the plant from a store so it is hard to find the purity and originality.

Reply: Thanks for the comment. We have performed the gene sequence analysis, and the results have been included in Supplementary Figure S1 of this revised paper. The sequence has also been submitted to GenBank (http://www.ncbi.nlm.nih.gov/nucleotide). The closest match resulting from the BLASTN search is Sequence ID: KY218782.1 (23 to 722 bp) with 99.43% identity. This description has been presented on Line 315-318.

Q4.  Please do explain why you have taken ethanol as a solvent for the extraction method?

Q5.   For the extraction the authors use ethanol that is proven mutagen and carcinogenic. So, they are advised to go for the residual solvent content of the extract.

Reply: Thanks for the comments. Ethanol (ethyl alcohol, C2H5OH) we used in this study was the food grade product, it is edible. This solvent has been widely used in the extraction of natural products and has been well-recognized as a safe additive in food preparation. The extract we used in this experiment was dry extract which had been vacuum concentrated and freeze-dried; so we believe the residual solvent in the extract was negligible. Such samples were further diluted when fed to fish, therefore, they are very safe for zebrafish.

Q6.  Please mention the manufacturer details of the standard diet and/or its composition if available.

Reply: Thanks for the comment. The label of the commercial fish feed product shows it contains at least 40.5% crude protein, 5.6% crude lipid, and 8% crude fiber. However, the label is rather simple and general, and we did not analyze the nutrients in detail, so we did not include in the paper.

Q7.   How the authors can determine the dose for pharmacological activity? If it is already mentioned somewhere in the literature. Please cite in the text.

Reply: Thanks for the comment. The doses used in this study were determined from the results of our preliminary experiments, in which the dosage tested were roughly between 0.01% - 20%. Three doses (0.2%, 1% and 10%) that were more suitable to exhibit the activity for in-depth investigation were then selected for this study.

Q8.  Please mention the manufacturer details of the incubator and other instruments used throughout the manuscript.

Reply: Thanks for the comment. The manufacturer information of the four instruments used in this study has been added to Materials and Methods section.

Q9.  Authors are strictly advised to first complete the full form of words then later they can use abbreviations (AlCl3 at line number 397).

Reply: Thanks for the comment. The complete full names of AlCl3 and NaOH have been added in the text (Line 414 and 415).

Q10.  The conclusion section is poorly written.

Reply: Thanks for the comment. The conclusion section has been rewritten and hope it is satisfactory and acceptable to the referee.

Q11.  The manuscript needs standard review.

Reply: Thanks for the comment.

While we do not fully understand the precise meaning of the term “standard review”, the aim of this study was to assess the application potential of AME in promoting sexual function. In our future follow-up study, we will explore in more detail the role of AME in increasing sperm number and its protective effect, as well as improving pregnancy rate. If it is to be made into a health food product, it will be prepared in accordance with government’s standard and regulations. Therefore, there are still many more in-depth and detailed studies that need to be explored. We hope that our reply has responded correctly to the referee's question.

We are very grateful to your helpful and constructive comments. Accordingly, we have carefully revised this manuscript as suggested. We have highlighted changes in the revised manuscript in different colors. Hopefully, the above changes and responses are satisfactory. Please do not hesitate to contact us if any ambiguities remain. We look forward to your favorable response.

With our best regards

Sincerely yours,

Chi-Chang Chang

Reviewer 2 Report

The manuscript: Effects of Abelmoschus manihot Flower Extract on Enhancing 2
Sexual Arousal and Reproductive Performance in Zebrafish, describe information about the effect of the target plant in the Zebrafish animal model related to sexual arousal and reproductive performance.

The authors mention that flavonoids are one of the main constituents of the flower extract, they suggest that these compounds may have the potential to protect sperm and improve pregnancy rate, and conduct the chemical analysis-oriented only to flavonoids. I wonder why only flavonoids, being that are a total extract and contain many compounds. This part needs to be justified to a better understanding.
to protect sperm and improve pregnancy rate

Author Response

Dear Referee,

Attached please find our revised manuscript (molecules-1650301) entitled “Effects of Abelmoschus manihot flower extract on enhancing sexual arousal and reproductive performance in zebrafish” which we wish to publish in Molecules journal. We thank you for your thoughtful comment, which greatly improve the present manuscript. Following your comment, we have carefully revised the manuscript, and listed our response as follows.

Q1.    The authors mention that flavonoids are one of the main constituents of the flower extract, they suggest that these compounds may have the potential to protect sperm and improve pregnancy rate, and conduct the chemical analysis-oriented only to flavonoids. I wonder why only flavonoids, being that are a total extract and contain many compounds. This part needs to be justified to a better understanding.

Reply:   Thanks for the comment.

Flower of A. manihot is a medicinal and edible natural material with many bioactivities and medicinal uses. More than 20 papers have attributed these bioactivities to its flavonoid composition. In light of this conclusion, we focused on the analysis of flavonoid components to show that the material used in this study is similar to those reported in the literature. Furthermore, as suggested by another referee, we have added the result of gene sequence analysis to this revised paper to confirm the identity of this plant.

As stated by the referee, the sample we used was a total extract and should contain many other compounds. Indeed, the active ingredients for sexual function promotion are not necessarily flavonoids. Therefore, we have modified the statement according to the reviewer's suggestion: “Since the total extract contains many other compounds in addition to flavonoids and polyphenols, whether these compounds exert any sexual function-promoting effect needs further investigations.” (Line 306-310).

We are very grateful to your helpful and constructive comment. Accordingly, we have carefully revised this manuscript as suggested. All changes are highlighted in different colors in the revised manuscript. Hopefully, these changes and responses are satisfactory. Please do not hesitate to contact us if any ambiguities remain. We look forward to your favorable response.

With our best regards

Sincerely yours,

Chi-Chang Chang